# Craniovertebral Junction Instability after Oncological Resection: A Narrative Review

**DOI:** 10.3390/diagnostics13081502

**Published:** 2023-04-21

**Authors:** Malte Ottenhausen, Elena Greco, Giacomo Bertolini, Andrea Gerosa, Salvatore Ippolito, Erik H. Middlebrooks, Graziano Serrao, Maria Grazia Bruzzone, Francesco Costa, Paolo Ferroli, Emanuele La Corte

**Affiliations:** 1Department of Neurological Surgery, University Medical Center Mainz, 55131 Mainz, Germany; 2Department of Radiology, Mayo Clinic, Jacksonville, FL 32224, USA; 3Head and Neck Department, Neurosurgery Division, Azienda Ospedaliero-Universitaria di Parma, 43126 Parma, Italy; 4Department of Neurosurgery, Mayo Clinic, Jacksonville, FL 32224, USA; 5Department of Health Sciences, San Paolo Medical School, Università Degli Studi di Milano, 20142 Milan, Italy; 6Department of Neuroradiology, Fondazione IRCCS Istituto Neurologico Carlo Besta, 20133 Milan, Italy; 7Department of Neurosurgery, Fondazione IRCCS Istituto Neurologico Carlo Besta, 20133 Milan, Italy

**Keywords:** atlanto-occipital joint, atlantoaxial fusion, spinal fusion, chordoma, endoscopic surgical procedure, skull base, joint instability

## Abstract

The craniovertebral junction (CVJ) is a complex transition area between the skull and cervical spine. Pathologies such as chordoma, chondrosarcoma and aneurysmal bone cysts may be encountered in this anatomical area and may predispose individuals to joint instability. An adequate clinical and radiological assessment is mandatory to predict any postoperative instability and the need for fixation. There is no common consensus on the need for, timing and setting of craniovertebral fixation techniques after a craniovertebral oncological surgery. The aim of the present review is to summarize the anatomy, biomechanics and pathology of the craniovertebral junction and to describe the available surgical approaches to and considerations of joint instability after craniovertebral tumor resections. Although a one-size-fits-all approach cannot encompass the extremely challenging pathologies encountered in the CVJ area, including the possible mechanical instability that is a consequence of oncological resections, the optimal surgical strategy (anterior vs posterior vs posterolateral) tailored to the patient’s needs can be assessed preoperatively in many instances. Preserving the intrinsic and extrinsic ligaments, principally the transverse ligament, and the bony structures, namely the C1 anterior arch and occipital condyle, ensures spinal stability in most of the cases. Conversely, in situations that require the removal of those structures, or in cases where they are disrupted by the tumor, a thorough clinical and radiological assessment is needed to timely detect any instability and to plan a surgical stabilization procedure. We hope that this review will help shed light on the current evidence and pave the way for future studies on this topic.

## 1. Introduction

The craniovertebral junction (CVJ) represents the most mobile and dynamic region of the entire axial skeleton. Phylogenetic evolution has developed an elegant osteoligamentous structure as an instrument for space exploration, and this allowed for the development of specific human eye–hand coordination movements. Numerous papers have focused on the description of the anterior, posterior and posterolateral approaches to CVJ tumors and the management of the tumor-associated CVJ instability [1,2,3,4,5]. 

However, a comprehensive review on this topic discussing the surgical anatomy, the different types of tumors harboring in this region, the radiological criteria for CVJ instability, the pros and cons of the different surgical approaches—including a dedicated focus on the instability-inducing features of each approach—and an updated overview over the fixation techniques and material is missing in the current literature. This narrative review aims to summarize the recent evidence in this field and provide a concise but practical overview on CVJ instability after oncological resection, with a special focus on the algorithms and fixation techniques currently used in the treatment of patients with tumor-induced CVJ instability. This paper was conducted according to the Scale for the Assessment of Narrative Review Articles (SANRA) guidelines [6]. 

### Search Strategy and Studies Selection

English articles were retrieved from PubMed and MEDLINE databases using pertinent search terms related to CVJ tumor and CVJ instability, including the search terms “craniovertebral junction”, “C0–C2”, “craniovertebral junction instability”, “craniovertebral junction AND endoscopic endonasal approach”, “craniovertebral junction AND posterior approach”, “transoral approach”, “atlanto-axial fusion” and “far lateral approach”. Original articles including clinical and anatomical studies, review articles, systematic reviews and meta-analyses were considered eligible. No specific time restriction was applied. A special focus was devoted to the latest evidence in light of the available state-of-the-art treatment. This review summarizes the syntheses of the data retrieved from relevant publications. 

## 2. Anatomy and Biomechanics

The CVJ represents the transition between the skull and the spine. It consists of the occipital bone (C0) and the first two cervical vertebrae, the atlas (C1) and axis (C2). These bones articulate and interact with each other through a series of capsular–ligamentous and muscular structures, which characterize the CVJ as the most mobile joint. It also contains the delicate central nervous system (CNS) cervical–medullary junction transition region [7,8].

### 2.1. The Occipital Bone

The occipital bone surrounds the foramen magnum and constitutes the floor and posterior wall of the posterior cranial fossa. It is divided into the squamous portion, the basilar portion and two condylar portions, which form the posterior, anterior and lateral margins of the foramen magnum, respectively. On the intracranial side, the medial surfaces of the occipital condyles have a tubercle into which they are inserted: the alar ligaments; the internal orifice of the anterior condylar canal (or hypoglossal canal) directed anteriorly and laterally, which gives passage to the hypoglossal nerve; and the internal orifice of the posterior condylar canal of transit for the condylar emissary vein of the sigmoid sinus. Superolateral to the hypoglossal canal and inferomedial to the petroclival fissure is the jugular tubercle, which is an oval-shaped prominence that forms the medial border of the jugular foramen. On the exocranial side, anteriorly and superiorly to the occipital condyle, the supracondylar groove can be distinguished, which is medial to the external orifice of the hypoglossal canal [9]. Posterolateral to the occipital condyle is the jugular process, which is quadrangular in shape and forms the posterior border of the jugular foramen. 

### 2.2. The Atlas

The atlas is the first of the vertebrae; it has no body and no spinous process. Instead of a body it has two lateral masses that join anteriorly in the anterior arch and posteriorly in the posterior arch. The anterior arch of the atlas represents 1/6 of the circumference of the inner ring and ventrally presents the anterior tubercle, whereby the anterior longitudinal ligament is inserted, and it posteriorly presents a concave facet that houses the odontoid process of the axis. The lateral masses represent 2/6 of the circumference of the internal ring of the atlas and are constituted superiorly by the superior articular facet with a concave shape that houses the ipsilateral convex occipital condyle, and they are also constituted inferiorly by the inferior articular facet, which relates to the superior joint of the axis. Medial to each lateral mass is a process where the transverse ligament attaches. Between the transverse process and the lateral mass is the transverse foramen, which encases the vertebral artery. The posterior arch represents 3/6 of the circumference of the internal ring of the atlas and has a groove on its upper surface where the horizontal section of the vertebral artery and the nerve root of C1 lie. It also has a posterior tubercle where the posterior longitudinal ligament is inserted.

### 2.3. The Axis

The axis, the second cervical vertebra, is characterized by the odontoid process, a cylindric-shaped prominence that originates from the body of the axis and is directed superiorly. Anteriorly, it has an articular facet that faces the articular facet of the anterior arch of the atlas, and posteriorly, it has a groove where it houses the transverse ligament. Furthermore, the apex of the tooth is the site of insertion of the alar and apical ligaments. The transverse processes and the superior and inferior articular facets detach from the body of the axis and articulate, respectively, with the atlas and the superior articular process of the third cervical vertebra. Lateral to the superior articular facet and medial to the transverse process is the transverse foramen through which the vertebral artery passes. The peduncles and laminae constitute the remaining circumference of the vertebral ring; the latter merge posteriorly and in the midline and form the spinous process of the axis.

### 2.4. The Joint Complex

The atlas is a bony ring which, by means of its lateral masses, joins the occipital bone to form a condylarthrosis, which allows the head to flex and extend, but it is not the site of rotational movements except by a few degrees. The atlas articulates with the axis at two points: the lateral atlantoaxial joint, which is between the articular processes present in the entire column, and the median atlantoaxial joint, in which all the rotational movements of the head take place. The anterior and posterior atlanto-occipital membranes contribute to the atlanto-occipital joint. The anterior atlanto-occipital membrane connects the anterior border of the foramen magnum to the anterior arch of the atlas, while the posterior atlanto-occipital membrane connects the posterior border of the foramen magnum to the posterior arch of the atlas. The lateral portion of the posterior atlanto-occipital membrane is detached and arches over the vertebral artery and the root of the first spinal nerve (C1). The dens of the axis articulate anteriorly with the facet joint connection of the posterior surface of the anterior arch of the atlas and posteriorly with the transverse ligament. The strong transverse ligament that is inserted via the two medial processes of the lateral masses is an important structure in the stabilization of the CVJ. Two longitudinal fibrous bundles branch off from the transverse ligament: the superior one, which is inserted on the basilar portion of the occipital bone between the tectorial membrane and the apical ligament, and the inferior one, which is inserted on the posterior surface of the body of the axis. The transverse ligament together with its two longitudinal bands is also known as the cruciform ligament. Posterior to the cruciform ligament and adjacent to the dura of the cervicomedullary junction, a thickened robust ligamentous structure called the tectorial membrane connects the axis to the basilar portion of the occipital bone. Anterior to the cruciform ligament are the alar ligaments and the apical ligament, which are firm structures that connect the tooth of the axis to the occipital bone. The anterior longitudinal and posterior longitudinal ligaments connect the anterior tubercle of the atlas to the anterior surface of the axis and the posterior tubercle of the atlas to the spinous process of the axis, respectively.

### 2.5. The Craniocervical Muscle Complex

The CVJ is surrounded by various muscular structures that participate in the stabilization and mobilization of the most dynamic joint of the axial skeleton. The osteoligamentous complex of the CVJ, which is unstable in itself, is stabilized by the craniocervical musculature, which exerts a clamping effect [10]. From the surface to the depths, the anatomical planes offer skin, subcutis and superficial or trapezius fascia, which continues laterally in the superficial cervical fascia. The nape muscles, all subfascial, form four planes. The foreground is occupied by the upper part of the trapezius muscle. The only second-plane muscle that has primary action on the CVJ is the splenius. The third plane includes three longitudinal muscles acting on the CVJ: the semispinalis capitis and the longus capitis. In the fourth deepest plane of the region, there are shorter segmental muscles: the two rectus muscles, the two oblique muscles, the spinal transversus and the interspinous and intertransverse segmental muscles.

The prevertebral region is essentially composed of that thin layer of fascia that covers the anterior face of the cervical spine and extends up to the basilar surface of the occiput and down to the first thoracic vertebra. Three planes make up the region, from front to back: the prevertebral fascia, the prevertebral muscles (longus and rectus anterior capitis, longus colli and anterior intertransverse muscle) and the anterior aspect of the vertebral bodies. 

### 2.6. Biomechanics Aspects

The complex of joints that contributes to the CVJ structure allows for greater freedom of the entire axial skeleton, as flexion–extension, lateral inclination, rotation and circumduction movements of the head are permitted. Flexion–extension movements with a range of motion (ROM—range of motion) of about 25° are mainly performed at the atlanto-occipital joints, to which is added a ROM of about 10–20° obtained at the atlanto-occipital axial joints [11]. Flexion is mainly constrained by the tension of the cruciform ligament, which guarantees a predental space <3 mm in adults and <5 mm in children; by the possible block that follows the contact of the dens of the axis and the occipital; and by the stabilizing action of the cruciform ligament [12,13]. Extension movements, on the other hand, are bound by the contact between the opisthion and the posterior arch of the atlas [14]. The tectorial membrane does not appear to perform a stabilizing function, but during extension movements, it plays the role of preventing the tooth from compressing the dura mater [15,16].

Lateral tilt movements are mainly performed at the level of the atlantoaxial joints with a ROM of about 6°, while the ROM at the level of the atlanto-occipital joints is around 3°. Regarding lateral movements of the head, the alar ligaments perform an important stabilizing action.

The rotational movements of the head are mainly carried out in the atlantoaxial joints with an average ROM of about 23–40° and in the atlanto-occipital joints with a ROM of about 2–8°. The rotational movements of the head are limited by the contact between the inferior articular facet of the atlas and the superior articular facet of the axis and by the secondary constraint to the tensioning of the ligaments the contralateral transversus and alar [10]. 

## 3. Pathological and Radiological Features of the CVJ Neoplasms

Neoplastic disorders primarily affecting the craniovertebral are rare; they account for 0.5% of all spinal tumors, are more common in adults than in children and include both a benign and malignant pathology (primary and secondary) [17,18]. Primary neoplasms of extradural origin include chordomas, nerve sheath tumors, paragangliomas, osseous benign neoplasms (fibrous dysplasia, osteoid osteoma, osteoblastoma, aneurysmal bone cyst, giant cell tumors, osteochondroma, chondroma and eosinophilic granuloma), osseus malignant neoplasms (chondrosarcoma, plasmacytoma or multiple myeloma, Ewing’s sarcoma) and metastasis from other primary tumors. Primary neoplasms of intradural extramedullary origin include meningiomas, nerve sheath tumors and congenital cystic lesions (arachnoid cyst, epidermoid cyst and neurenteric cyst) [17,18,19]. Intradural pathologies that can affect the bulbomedullary junction are divided into extramedullary lesions (meningioma, schwannoma, neuroenteric cyst) and intramedullary lesions (glioma, ependymoma, metastasis) [9,20]. Secondary tumors include the extension of local malignancies (such as nasopharyngeal malignancy and rhabdomyosarcomas) and metastatic tumors [21]. Each tumor has different patterns of bone damage and the involvement of neurovascular structures, but no single neurological symptom or sign is pathognomonic for a specific type of lesion in this location [17]. Because of the large diameter of the spinal canal at the CVJ, symptoms arise only after a tumor has achieved a very large size or through osseous destruction and tumor-induced instability [4]. 

The most common presentation is refractory pain, which occurs in 70% of patients; this occurs in the suboccipital region and is usually aggravated by neck and head motions [4,22]. Patients can have a wide constellation of motor and sensory signs and symptoms including paresthesias or dysesthesias of the face and extremities, spastic weakness and cruciate paralysis [23,24]. A fluctuating neurological course with transient symptoms and false localizing signs that lead to an erroneous diagnosis are often reported due to the spinal cord compression and rotation with contralateral traction in this anatomically complex region with the decussation of the sensory and motor tracts [25]. The other mechanisms involved are anterior spinal artery compression, vertebrobasilar insufficiency (VBI), venous stagnation with spinal cord edema and hydrocephalus/hydromyelia secondary to cerebrospinal fluid (CSF) obstruction [14,26,27,28]. 

Other specific symptoms depend on the location, which reflects the involvement of neighboring neurovascular structures. When the tumor involves the intracranial region, tinnitus, vertigo, hearing loss, dysphagia, slurred speech and episodes of aspiration can be present due to the compression of the lower cranial nerves (vestibulocochlear, vagus, glossopharyngeal and hypoglossal). Occasionally, cerebellar symptoms can occur [29,30]. Straddle lesions at the level of the foramen magnum usually cause high cervical myelopathy. Restless legs syndrome has also been described in these patients [31]. Patients with high cervical lesions can have torticollis and weakness of the trapezius and sternocleidomastoid muscles due to the involvement of the spinal accessory nerve [32]. These lesions have also been associated with an abnormal cold sensation in the lower limbs and with a “dissociated sensory loss” consisting of a selective loss of pain and temperature while fine touch and proprioception are preserved.

The diagnosis of CVJ neoplasms has been facilitated through modern neurodiagnostic imaging techniques including flexion/extension radiography of the craniocervical region, Magnetic Resonance Imaging (MRI) with/without contrast, MR Angiography (MRA), MR Venography (MRV), Computed Tomography (CT) and three-dimensional CT Angiography (CTA) [33]. The evaluation of the extent and direction of the encroachment, the bone destruction, the neurovascular involvement, the associated edema/syringomyelia/hydrocephalus and the craniovertebral stability allow us to determine the nature of the lesion and to select the most appropriate surgical strategy [17,19]. 

Chordomas, meningiomas and nerve sheath tumors (schwannomas and neurofibromas) represent the most common primary neoplasms [33,34]. 

### 3.1. Chordomas

Chordoma, although rare, represents the most common histotype of primary malignant tumors found in this region (Figure 1). Chordomas originate from embryonic remnants of the primitive notochord along the axial skeleton. Chordomas usually occur in adults with a peak incidence between ages 50 and 60, and prognosis is typically poor with a 10-year survival rate of approximately 40% [35]. Less than 5% of chordomas arise in children, and in these cases, they show a more aggressive behavior with a greater incidence of metastasis and mortality, especially when the children are younger than 5 years old [36]. The classic midline clival chordoma is associated with spheno-occipital synchondrosis and can infiltrate the jugular fossa and foramen magnum, which leads to the erosion of the upper cervical vertebrae. The gross entire or nearly total excision of chordomas of the cranial base should be performed whenever possible because of their propensity to recur locally and their low likelihood to metastasize [37,38]. Stereotactic radiosurgery and proton beam therapy have emerged as a valuable adjunct to surgery, and patients benefit from craniovertebral stabilization [39,40].

Chordomas present as firm, pinkish gray, gelatinous masses composed of a myxoid matrix containing necrotic and hemorrhagic areas, calcifications and sequestered bone fragments. Intrinsic tumor vascularity is usually not pronounced. Chordomas have been divided into classic chordomas, chondroid chordomas and dedifferentiated chordoma. Classic chordomas account for 80% to 85% of all chordomas [41]. Dedifferentiated chordomas have a sarcomatoid appearance with large areas of necrosis, and they tend to be aggressive and have the worst prognosis [42]. 

On CT, chordomas appear as well-defined expansile soft tissue masses that arise from the clivus with associated destructive lytic lesions and occasionally marginal sclerosis [43]. They appear as hypoattenuating heterogeneous lesions with areas of necrosis, hemorrhage and calcifications. Chordomas are iso/ipo-intense on T1-weighted MRI, with small hyperintense foci relative to intratumoral calcifications, hemorrhages or mucus pools. After contrast material injections, a honeycomb pattern of enhancement with intratumoral areas with a low signal intensity is typical. T2-weighed MRIs reveal a bright hyperintensity with heterogeneous hypointensities [38,44,45]. The “thumb sign” is a radiological finding that can be visible when chordomas project posteriorly at the midline and indent the pons [46].

### 3.2. Meningiomas

Meningiomas are arachnoid cap-cells-derived tumors that represent the second most common tumors in the CVJ region. Meningiomas affect women in about 80% of cases and exhibit estrogen and progesterone sensitivity. In children, they have the same incidence among boys and girls, and they typically develop in association with neurofibromatosis (NF) type II. Meningiomas are exclusively intradural in almost all cases but can occasionally have an extradural extension. Meningiomas located in the CVJ region primarily attach to the foramen magnum’s anterior rim, and they can also infiltrate the vertebrobasilar region, which involves the cranial nerves IX through XII and the cervical nerve roots [47]. The World Health Organization (WHO) classify meningiomas in three grades: grades I meningiomas represent 80% of all cases and have a benign behavior. Grade II (atypical) and grade III (anaplastic) meningiomas are rare in the CVJ region and tend to grow more quickly, with grade III meningiomas having a malignant behavior [48,49]. Among the 15 histo- and cytomorphological variants of meningiomas, various subtypes of meningiomas have been found in this location, including meningothelial (most common), xanthomatous, fibrous, lymphoplasmacytic-rich, transitional and clear cell [50,51]. Clear cell meningiomas have the potential to behave aggressively and metastasize through the cerebrospinal fluid as well as to recur locally [52]. Typically, meningiomas are well circumscribed, appear isodense or moderately hyperdense on CT scans and isointense on T1- and T2-weighted MRIs and show a homogeneous enhancement after a contrast injection on both CT and MRI scans [19]. Gadolinium-enhanced sequences help to define the relationship with the brainstem and the other neurovascular structures and a broad-based dural attachment with a “dural tail” often visible. The imaging might vary significantly according to the specific subtype [53]. The “ginkgo leaf” sign has been described in the literature regarding the differential diagnosis of lateral or ventrolateral CVJ meningiomas and neurogenic tumors [54].

### 3.3. Spinal Nerve Sheath Tumors 

Nerve sheath tumors of the CVJ are rare and include schwannomas and neurofibromas that arise from the lower cranial nerves, in particular the hypoglossal nerve, and upper cervical spinal nerves including C1 and C2 [55]. Their peak incidence is the fourth decade of life without sex predilection, and they are benign in 90% of cases. Malignant peripheral nerve sheath tumors (MPNSTs) are a rare subtype of nerve sheath tumors that show a high grade malignant behavior [56]. Tumors can be intradural extramedullary or entirely extradural, and they usually arise in a dorsal location. If they become large, they may either align with the long axis of the spinal cord and generate fusiform masses that can extend over multiple levels, or they may protrude out of the neural foramen with an intracanal and a paravertebral component (called dumbbell nerve sheath tumors) [57,58]. Multiple upper cervical nerve sheath tumors characteristically occur in cases of neurofibromatosis. Schwannomas are associated with NF type II while plexiform neurofibromas and MPNSTs are associated with NF type I [59,60]. 

Radiographically, neurofibromas and schwannomas appear well defined and can be easily confused, but while neurofibromas typically have a fusiform shape, schwannomas are commonly rounded. On a CT scan, the density can vary from hypodense to slightly hyperdense, and it can be possible to observe a widening of the neural exit foramina, bone erosion and vertebral body scalloping. On MRIs, the signal characteristics include isointensity in T1 and hyperintensity with mixed central signals in T2. T1 gadolinium enhancements are homogeneously intense in schwannomas and heterogeneous in areas of low signals in neurofibromas [61,62]. Features such as hemorrhages, thrombosis, sinusoidal dilatations, cyst development, fatty degeneration and the displacement of the nerve roots are more specific of schwannomas and are unusually seen in neurofibromas, which show a more symmetrical growth with the encasement of the nerve roots [63].

### 3.4. Chondrosarcomas 

Chondrosarcomas originate from primitive mesenchymal cells or from embryonal rests of the cartilaginous matrix and represent the main differential diagnosis of chordoma in the CVJ region. Differently from chondromas, they rarely occur in the midline and do not express brachyury [64,65]. 

### 3.5. Secondary Tumors of the CVJ

Secondary tumors of the CVJ are primarily plasmacytomas and metastases. Plasmacytomas of the CVJ region are monoclonal plasma cell neoplasms that arise from the nasopharyngeal submucosal tissue (extramedullary plasmacytoma) or the marrow space of the clivus or cervical vertebrae (solitary plasmacytoma of bone) (Figure 2). Plasmacytomas show a homogeneous signal intensity on T1- and T2-weighted MR images [66]. Breast carcinoma is the most common primary site of metastasis in the CVJ, and it is reported to be the site for one third of all cases [67]. Other primary tumors include thyroid, lung, renal and prostate carcinomas. The signal intensity and enhancement pattern on MR images may reflect the cell type of the original malignancy and may be associated with necrosis or hemorrhages [21].

## 4. Radiological Criteria of CVJ Instability

The symptoms induced by CVJ instability may range from neck pain and restricted neck movements to sensory and motor abnormalities and gait instability [68]. The concept of CVJ instability is based on both bony abnormalities and on excessive laxity/loss of insertion of the atlanto-occipital and atlantoaxial ligaments. The combination of a hypoplastic dens in association with anomalies of the posterior arch of the atlas increases the risk for craniocervical instability. Injuries to the anterior atlanto-occipital membrane, the apical and alar ligaments, the cruciate ligaments and the tectorial membrane result in instability since these ligaments provide most of the stability to the atlanto-occipital joint [69]. Thin-section multidetector dynamic CTs of the craniocervical region with sagittal and coronal sections, lateral radiographs of flexion/extension and dynamic MRIs of flexion/extension are the studies of choice to evaluate the stability of the CVJ [70]. The radiological measurements most commonly used for the diagnosis of CVJ instability include (Figure 3): The Clivoaxial Angle (CXA), which is the angle between the clivus line and the posterior axial line, examines the brainstem deformity induced by the odontoid process. A CXA of 135 degrees or less is considered “potentially pathological” [71].The Grabb–Oakes line, which is the perpendicular distance from the dura to the line drawn from the basion to the posterior inferior edge of the C2 vertebra. It is a measure of the encroachment of the odontoid process into the upper spinal canal (basilar invagination) and investigates ventral brainstem compression. A measurement ≥9 mm is considered pathological [68].The Basion–Dens Interval (BDI) measures the vertical distance between the basion and the dens and is considered pathological if ≥10 mm [72].The Basion–Axial Interval (BAI) is the distance from the tip of the basion to the posterior axial line and is pathological if ≥12 mm [72].The translational BAI and translational BDI are the change in mm of the BAI and BDI between the flexion and extension positions of the head [68].The Atlantodental Interval (ADI) is the distance between the posterior surface of the anterior atlas ring and the anterior surface of the odontoid process. An ADI >5 mm in adults and >4 mm in children is an indication for surgery [73].The Condyle–C1 interval (CC1) measures the distance between the occipital condyle and C1 at four equidistant points and is pathological in children if >4 mm, with a high diagnostic accuracy [73].The Powers ratio is calculated by measuring the distance between the basion and the posterior arch of the atlas and then dividing it by the distance between the opisthion and the anterior arch of the atlas [74].

The Traynelis classification classifies occipitocervical dissociation patterns into three types according to the direction of dislocation of the occiput relative to C1: type I (anterior displacement), type II (superior–inferior displacement) and type III (posterior displacement) [75]. A limitation of this classification system is that rotatory or coronal malalignments are not taken into consideration. The Harborview classification described three levels of severity of CVJ instability. In Stage 1, there is sufficient preservation of ligamentous integrity, and a conservative treatment is indicated. In Stage 2, the craniocervical alignment is within 2 mm of normal. In Stage 3, there is a craniocervical malalignment of >2 mm, which requires internal fixation [76]. Horn et al. proposed another grading scale based on both CT and MRI scans. Grade I is defined by normal CT measurements (described above) but moderately abnormal MRI findings, and in these cases, a conservative treatment is indicated. Grade II is characterized by minimal abnormalities in the CT measurements but grossly abnormal MRI findings regarding the atlanto-occipital joints, tectorial membrane, alar ligaments or cruciate ligaments. In these cases, a surgical intervention is indicated [77].

Mechanical instability of the CVJ may result from osteolytic destruction and dislocation induced by the tumor or from the surgical and radiation treatments. However, the CVJ instability in these cases rarely follows the injury patterns seen in the trauma population, and specific guidelines for the neoplastic setting in this region are not yet defined [4]. The Spinal Instability Neoplastic Score (SINS) was developed by the Spine Oncology Study Group (SOSG) in 2010 and assesses and scores six variables: location, characterization of pain (from 1 to 3), type of bone lesion (lytic, mixed, blastic), radiographic spinal alignment (normal, deformity, subluxation), degree of vertebral body destruction (insolvent, degree of collapse) and involvement of posterolateral structures (unilateral, bilateral). The total score can range from 0 to 18, and instability is defined by a score of ≥13. However, a score between 7 and 12, which indicates a potentially unstable lesion, is an indication for surgical consultation [78]. A lytic destruction or resection of 70% of a unilateral condyle, 50% of bilateral condyles or extensive removal of the posterior elements and facets are also suggested in the literature as indications for occipitocervical fixation (OCF) [34]. 

## 5. Surgical Approaches to the CVJ

Several approaches are available in the surgical armamentarium to deal with a lesion harboring into or near the CVJ. The selection of the optimal surgical strategy is paramount for the success of the operation, as this maximizes the chances of achieving the surgical goals and minimizing the surgical related morbidity. In the table reported below (Table 1), we briefly summarize the pros and cons of each surgical approach. 

For a better overview of each approach, the reader is referred to the dedicated sections that follow.

## 6. Anterior Surgical Approaches to the CVJ

### 6.1. Transoral Approach

The transoral (or buccopharyngeal) approach is one of the most commonly used surgical approaches to decompress the craniovertebral junction affected by ventral, irreducible and extradural pathological lesions [79,80,81]. This approach has the ability to expose the anterior region of the foramen magnum from the basilar portion of the occipital bone to the vertebral body of C3 [81,82]. It was first described by Kanavel in 1917 who used it to remove a projectile located at the craniocervical junction. Subsequently, the microsurgical technique was popularized and refined by some expert neurosurgeons such as Alan Crockard and Arnold Menezes. From the preoperative point of view, it is necessary that patients who are candidates for this intervention carry out any drainage of the oral cavity to minimize the oral bacterial load [79,80,81]. It is also advisable to check the functionality of the mixed nerves and possibly proceed with a prophylactic tracheotomy. The maximum opening of the oral cavity should be promptly investigated especially in pediatric patients or in patients with macroglossia (down syndrome), and in such cases, an extended approach may be required (such as a mandibulotomy or glossotomy) [82]. The patient is positioned supine with the head extended and intubated via the orotracheal or tracheal route (this process is video-assisted, especially in patients with craniocervical instability), and gauzes are placed in the laryngopharynx to prevent the passage of blood into the stomach. After disinfecting the oral cavity with chlorhexidine and performing intravenous antibiotic therapy, the oral retractor (e.g., Crockard or Spetzler–Sonntag) is positioned. Hsu et al. recommend releasing the tongue from retraction at an interval of thirty minutes to avoid the risk of lingual edema from venolymphatic compression [80]. This buccopharyngeal approach can be associated with extensions to obtain a wider surgical window and in selected cases with a reduced buccal opening (“extended transoral approach”) [82,83,84,85].

Performing a mandibulotomy facilitates the surgical exposure of the region, reduces the operative distances and increases the angle of the craniocaudal and lateral exposure, which allows for optimal control from the middle third of the clivus to the C2–C3 intervertebral space. Performing a mandibuloglossotomy increases the sagittal exposure from the upper third of the clivus to the C4–C5 intervertebral space together with the reduction in the operative distance.

A palatotomy increases the rostral exposure to the upper third of the clivus without changing the operative distance or caudal or lateral exposure. A Le Fort I maxillotomy allows for lower jaw and hard palate mobilization, which allows for greater rostral control up to the sphenoid sinus and greater lateral control. The obstacle in the caudal exposure of this approach can be overcome by combining a median split of the maxillary bones, the hard palate and the soft palate [82,83]. Although, these approaches and transoral variants involve a decrease in the distance of the working canal, and increased surgical exposure is often associated with increased complications in the postoperative period [1,86]. A velopalatine incision and functional Passavant ring infringement may increase the risk of velopalatine insufficiency, whose symptoms and clinical signs include a nasal voice, dysphagia and fluid regurgitation. The incision of the tongue can lead to impaired motility of the tongue and therefore dysarthria. These procedures are also associated with an increase in the intubation period and the return to independent oral nutrition [81]. An important complication of this approach is the risk of liquorrhea; in the presence of this, it is necessary to resort to an aggressive repair treatment to avoid the onset of infectious conditions, which can lead to devastating consequences [80]. A lateral surgical exposure of the joint could lead to damage to the four cerebral arterial axes (carotid arteries and vertebral arteries), especially in the presence of anatomical variants, such as the retropharyngeal position of the carotid arteries; a correct preoperative and intraoperative radiological classification is therefore necessary with the use of a neuronavigator.

### 6.2. Transoral Endoscopic Approach

The relevance of the comorbidities associated with the transoral approach and its extensions has led the scientific community to identify new minimally invasive ways to manage the craniovertebral junction pathology [2,87,88]. The advancement of endoscopic techniques has allowed surgeons to perform minimally invasive procedures in the face of more aggressive and disabling traditional interventions at the craniovertebral junction. Frempong-Boadu et al. demonstrated for the first time the feasibility of the endoscope-assisted transoral approach for the treatment of joint anomalies [2]. Since then, there have been several anatomical studies and case reports. The use of the endoscopic technique has the advantage of a better visualization of the operative field while minimizing surgical exposure. As previously illustrated, some extended variations can be employed to achieve greater rostral control of the clivus. The transoral use of the endoscope has allowed us to spare the incision of the hard palate together with the other bone structures, which guarantees a rostral approach up to the spheno-occipital junction [87,88].

### 6.3. Endoscopic Endonasal Approach

The endoscopic endonasal approach (EEA) to the craniovertebral junction (CVJ) is a minimally invasive surgical technique that allows for access to the CVJ through the nasal cavity (Figure 4). 

This approach has been used to treat a variety of conditions affecting the CVJ, including tumors, congenital anomalies and trauma. One of the advantages of the EEA is that it avoids the need for a large incision in the neck or skull, which can lead to less pain, a faster recovery and improved cosmetic outcomes [89,90,91]. Several surgical advantages have been attributed to the EEA in comparison to classical craniotomy or transfacial microsurgical techniques, and they have showed a reduced rate of morbidity and mortality [20,92]. In 2005, Kassam was the first neurosurgeon to use this technique to perform an odontoidectomy [93]. Subsequently, the pioneering work of the Pittsburgh group has been followed by numerous case reports and clinical series that all showed a reduction in the mortality rate and morbidity compared to the classic transoral approach [94,95,96,97]. The main advantages of the EEA to the craniovertebral junction derive from the surgical angle of attack and the surgical incision located at the level of the nasopharynx rather than the oropharynx. The nose and paranasal sinuses provide excellent rostral access and a rostrocaudal angle of attack for the treatment of most pathologies affecting this anatomical region; this is particularly true in patients with basilar invagination or when the odontoid has ascended upwards [98].

The foramen magnum and the superior portion of the cervical spine are located immediately behind the nasopharynx, which can be reached easily and directly through the nasal corridor. The nasopharynx has a significantly reduced proportion of virulent bacterial flora and neural plexuses that coordinate swallowing relative to the oropharynx [99]. The palate is never affected by the approach since the rostrocaudal angle of attack is parallel to that of the soft palate, and thus it does not require its transgression. The angle of approach and the caudal limit of access of the EEA to the CVJ is determined by the rhinopalatine line [100,101]. The lateral exposure of the EEA is limited by the parapharyngeal internal carotids and the jugular foramen. The transnasopharynx–transodontoid approach is an extension of the transclival approach. This approach can, however, be performed independently with preservation of the clivus since, in most cases, the pathology is confined to the level of the cervical spine; moreover, the exposure of the sphenoid sinus is not always required [102,103,104]. Furthermore, the possibility of stabilizing the CVJ through the use of transarticular screws positioned endoscopically via the endonasal have been described in anatomical studies [105,106]. In some situations, lateral exposure at the level of the foramen magnum is required [9]. This is particularly true in conditions affecting the occipital condyle and extending into the jugular foramen. In these situations, it is advisable to provide an appropriate lateral surgical window by removing the occipital condyle and partially resectioning the Eustachian tube. The reaming process of the occipital condyle can be performed until the hypoglossal canal is exposed. Careful neurophysiological monitoring of the cranial nerves is therefore essential to prevent the risk of deficits and to maintain the function of the hypoglossal nerve intact. If the medial occipital condylectomy is unilateral, by keeping the contralateral alar ligament intact, there is no risk of craniovertebral instability. 

### 6.4. Instability after EEA

However, one of the potential complications of the transclival–transodontoid EEA is the postoperative instability of the CVJ (Figure 5). 

This can occur when the stability of the CVJ is compromised by the surgical procedure or preoperatively by tumor growth, which leads to symptoms such as neck pain, weakness and spinal cord compression. The risk of postoperative instability can be reduced by careful preoperative planning, proper surgical techniques and appropriate postoperative management [107]. There are several different techniques that have been used to prevent instability after the EEA. These include the use of internal fixation devices such as screws and rods in the pre- or immediate postoperative course, as well as the use of external fixation devices such as halo traction. There have been several studies published in recent years that have investigated the incidence and management of postoperative instability after applying the EEA for CVJ tumors, with a focus on condyle resection. An anatomical study showed that a lower-third clivectomy and unilateral anterior condylectomy through an EEA can cause progressive hypermobility at the CVJ. On the basis of biomechanical criteria, OCF is indicated for patients who undergo a > 75% anterior condylectomy [108]. Kooshkabadi et al. evaluated the incidence of postoperative instability after an EEA for CVJ tumors. The study included 212 patients who underwent an EEA for lower clivus lesions, and they found that around 3.3% of the patients required a fixation. They showed that an EEA resection greater than 75% of the occipital condyle significantly increased the risk of CVJ instability, which required subsequent fixation. The degree of the condyle resection was a significant factor that predisposed it to the occipitocervical instability [109]. There are also some reports, mainly related to the EEA to CVJ abnormalities and basilar invagination, on the preservation of the anterior C1 arch that avoids the need for posterior fixation with the aim of preserving the rotational movement at the C0–C2 segment and reducing the risk of a subaxial instability development [98,110,111]. Overall, these studies suggest that the entity of condyle resections, C1 anterior arch and transverse ligament preservation while using the EEA on CVJ tumors may represent two significant factors that are related to the risk of postoperative instability [107,112]. However, the evidence is not entirely consistent, and further studies with larger patient populations and longer follow-ups are needed to better understand the risk. The most consistent risk factor identified across these studies is the size and location of the tumors, specifically tumors located in the upper cervical spine and tumors with a wide base. Other factors that have been identified as risk factors for postoperative instability include the degree of resection, the surgical approach and the reconstruction methods used.

## 7. Posterior Surgical Approaches to the CVJ

The posterior and posterolateral approaches represent two further operative techniques for the treatment of CVJ primary and metastatic neoplastic diseases available in the neurosurgical armamentarium (Figure 6).

The main variables determining the selection of the surgical approach are related to the tumor pathology, i.e., primary vs metastatic disease; anatomy, i.e., tumor location and involvement of the adjacent structures; and biomechanical considerations.

In the subsequent paragraph, we will briefly discuss the oncological and anatomical concepts guiding the selection of the posterior and posterolateral approach around the CVJ followed by a dedicated focus on the treatment of CVJ instability related to the aforementioned surgical approaches.

### 7.1. Type of Tumor and Aim of Surgery

Oncological lesions involving the CVJ can be generally classified as primary lesions, including bone, vascular or metastatic lesions according to their origin site. This first characterization is rather simple but extremely important as it is capable of providing indications about the aim of the surgery and, consequently, the selection of the surgical approach. 

Essentially, the surgical goal for primary tumors is focused on the resection of the lesion to cure the disease and/or improve the patient’s survival. Conversely, the aim for metastatic diseases is a palliative surgery with a functional purpose; relief of the spinal cord compression, acknowledged by the term separation surgery, is achieved through a decompressive procedure followed by subsequent safer radiation treatment to improve the neurological status and quality of life of the patient [113,114].

### 7.2. Approach Selection and Anatomical Considerations

The oncologic purpose significantly influences the surgical strategy in terms of the approach selection. Although no clear consensus has been reached regarding the recommended surgical approach—anterior, posterolateral or posterior—in the case of primary tumors, the Weinstein–Boriani–Biagini grading system and the anatomical location of the tumor with respect to the surrounding neurovascular structures still seem to be the most valuable indicators/factors to guide the selection of the surgical approach [1,114,115,116,117]. Instead, for the treatment of spinal metastasis, the posterior approach was found to be the most frequently selected surgical route in a recent meta-analysis by Fehlings et al. [22], whereby it was found to be used in about 75% of the cases. This choice is mainly dictated by biomechanical considerations regarding the stability of the CVJ, of which we will discuss more extensively in a following dedicated section. The anatomical location and the spatial relationships of the lesion with the surrounding neurovascular and bony structures represent a further crucial variable that guide the surgical strategy. In fact, several factors should be considered for the selection of the optimal approach. In the literature, the size of the tumor and its laterality (dorsal midline, posterolateral, lateral, anterolateral), relationship with neurovascular structures (e.g., vertebral artery and cranial nerves), degree of paraspinal extension, invasion of the surrounding tissues (e.g., bony erosion and/or tumor-induced bone fractures), entity of bony resection needed for the removal of the lesion and the necessity to relieve spinal cord compression are only a few of the anatomical factors tailoring the surgical approach [67,117].

### 7.3. Spinal Instability

Spinal instability, defined by the Spine Oncology Study Group (SOSG) as a “loss of spinal integrity as a result of a neoplastic process that is associated with movement-related pain, symptomatic or progressive deformity, and/or neural compromise under physiologic loads”, represents a crucial factor in the evaluation and management of a CVJ lesion [78].

According to the cause/etiology of instability, we can differentiate this condition in primary instability in the case that it is determined by the lesion itself, or secondary instability in the case that it is determined by a surgical maneuver. 

Regarding the primary instability, one-size-fits-all criteria for the assessment of spinal instability cannot be found in literature; however, the need for the establishment of standardized and common defining criteria was critical in clinical practice. In fact, as outlined by the NOMS (neurologic, oncologic, mechanic and systemic) framework, a decision framework that provides a comprehensive assessment for the management of patients affected by spinal metastasis based on four domains, mechanical instability is an independent surgical indication regardless of the neurological or oncological status [118]. 

Despite a specific scoring system that is able to predict spinal instability at the CVJ level, back in the day, the Denis three-column system was used as the reference method to assess instability [119].

Nowadays, the SINS, a consensus- and evidence-based score, is able to provide a stability assessment based on clinical and radiological features, and it represents the reference standard to determine instability and a precious allay in the decision-making strategy and when planning the tailored treatment [78]. The location of the lesion, presence of pain, type of bony lesions, degree of spinal alignment on radiographic imaging, entity of vertebral body collapse and laterality of the posterolateral spinal elements’ involvement were recognized as stability indicators, with a specific value according to each component of the score. In light of the final score, a degree of instability can be assessed, and this information can be used to guide the patient care with a high rate of sensitivity (96%) and specificity (80%) [67].

Conversely, secondary instability is determined by the biomechanical alteration given by the surgical maneuvers required to remove the tumor, according to the location, nature and size of the lesion as well as the need of the bony resection to reach and expose the neoplasm.

With regard to iatrogenic instability related to CVJ oncological resection trough posterior and posterolateral approaches, different instability-inducing approach-related mechanisms can be found in the literature. In the next paragraph, the current findings on this topic will be briefly summarized, with a special focus on the surgical approach, which highlights the risk factors associated with secondary CVJ instability.

### 7.4. Posterior Approach

The posterior midline approach represents a widely used approach for the removal of CVJ and cervical spinal cord tumors or performing a CVJ or cervical fusion with combined anterior–posterior approaches, with the aim of restoring spinal stability after a neoplasm resection (e.g., clival chordoma or metastatic disease). Intradural extramedullary tumors—e.g., meningiomas and schwannomas—and intradural intramedullary tumors—e.g., ependymomas—represent common entities encountered in this region, and they generally require a dorsal approach for their removal. Although a single-level laminectomy is generally considered safe for the maintenance of stability, the removal of the associated facet joint seems to be associated with spinal instability. In this regard, Sciubba et al., in light of previous evidence, considered that a unilateral facet compromises ≥50% or a bilateral facet compromises ≥33% as an indication for fixation [120,121,122]. Similarly, in the work of Jiang et al., the authors focused on the removal of CVJ extramedullary tumors, and fusion was recommended for the tumors that required a C1 facetectomy [123]. Therefore, despite the fact that the actual level of evidence is based mainly on case series and biomechanical analyses, occipitocervical fusion should be recommended in the case of a laminectomy and facetectomy for the removal of extramedullary lesions. Conversely, the indications for cervical spine fixations for multilevel laminectomies have not reached a unanimous consensus. Katsumi et al. identified several risk factors involved in the development of cervical spine instability, including the number of removed laminas (less than four vs more than four), C2 laminectomy and facet joint destruction [121]. On the other hand, in a recent case series reporting the Karolinska experience, Tatter et al. suggested that prophylactic posterior fixation could be avoided for “short” laminectomies (mean laminectomy range 2.4 ± 1.0 levels) with only a 2.4% delayed posterior fixation rate (2/84 patients) [124]. Nevertheless, a laminectomy of C2 and laminectomy of C3 were identified as risk factors related to an increase in the radiologic kyphotic angle. Finally, a contemporary systematic review and meta-analysis on this topic performed by Noh et al. reported an extremely wide incidence range in terms of deformity and instability after cervical spinal cord tumor resection in the adult population—0–41.7% and 0–20.5%, respectively [125]. Younger age, C2 laminectomy and a higher laminectomy level were identified as the associated risk factors. Additionally, Avila et al. tried to shed some light on this topic and provided a summary of the literature-derived fusion criteria [126]. The identified preoperative (first to sixth criteria), intraoperative (sixth criteria) and postoperative criteria (seventh criteria) that may help the surgeon in the decision process are the following: (1) spine deformity before surgery, (2) surgery involving three or more vertebral levels, (3) patients younger than 36 years old, (4) surgery that crosses a spinal transitional junction, (5) laminectomy of C2 vertebrae, (6) removal of 50% or more of the facets joints and (7) persistence of neck/back pain or failure of conservative management after 1 year.

### 7.5. Posterolateral Approach 

The posterolateral approach is part of the neurosurgical armamentarium for the removal of the lesions located in the lower part of the clivus, foramen magnum and the upper cervical spine [1,127]. The far lateral approach, in its transcondylar, supracondylar and paracondylar variants, is the most frequently used approach to treat lesions arising in the aforementioned anatomical locations such as meningiomas, schwannomas of the lower cranial nerves, Ewing sarcomas, etc. However, although these approaches are very useful, they may determine/create instability as a consequence of the bony resection required to expose the lesion. Apart from the considerations related to the removal of the C1 posterior arch and facetectomy, discussed in the previous point, the occipital condyle is the anatomical structure that deserves a special focus for its biomechanical and stability implications on the CVJ. Despite the fact that it is generally accepted that an extensive removal of the condyle will cause CVJ instability, the actual amount of removal capable of causing instability is still a matter of debate. Mazur et al., in a biomechanical analysis study, observed a significant biomechanical alteration when the extent of the condylar resection reached 29%, with a 5% increase in the range of motion (rom) of the flexion–extension movement, a 5% decrease in the stiffness of the flexion–extension movement and a 10% rom increase during the axial rotation movement [128]. Another biomechanical study, conducted by Vishteh et al., found different results, with an significant increase in the C0–C1 mobility only after the removal of 50% of the condylar mass, with a 15% flexion–extension, 41% lateral bending and 28% axial rotation increase [129]. Notably, with the application of the joint-sparing condylectomy technique as proposed by Kshettry et al., the cut-off value to induce biomechanical instability was reached only after a 75% condyle removal, with a significant increase in the rom (+28%) during lateral bending [130]. Finally, the clinical study performed by Shiban et al. surprisingly confuted the results obtained by the previous biomechanical analysis, whereby they showed that even the removal of the condyle superior to 75% was not associated with a clinically evident CVJ instability [131]. In light of those results, we endorse the concept that the decision for fixation should not be based only on dogmatic radiological criteria but in accordance with clinical status on an individual basis. In fact, the occipitocervical fixation procedure is not exempt from risks–including vascular and neural lesions—and therefore should be reserved for selected patients. 

## 8. Principles of Fixation Techniques and Materials

Tumors of the CVJ can either present with mechanical instability due to the tumor itself, or the CVJ can be destabilized by surgery or radiotherapy if relevant structures are resected or damaged. As mentioned in detail before, the CVJ is a complex anatomic region with specific structures that must be considered when assessing stability and planning the operation. In particular, the juxtaposition of neural structures and vessels presents a surgical challenge and often requires extensive bone removal to protect these structures.

### 8.1. Indication for Stabilization

A very useful tool to assess mechanical instability in the case of tumors is the SINS, as the SINS allows tumors to be categorized into three categories: stable, possibly unstable and unstable based on radiographic and clinical data. The SINS has demonstrated inter- and intraobserver reliability [26]. Due to the relatively wide spinal canal at the CVJ, the majority of patients become symptomatic with movement associated pain rather than neurological deficits due to spinal cord compression. Especially in “possibly unstable” cases, clinical findings regarding neck pain and radiographic findings, such as compromised ligaments, fractures or subluxation on dynamic radiographs determine the indication for fixation. The histopathological diagnosis of the tumor is not a factor that is regularly taken into consideration, although lytic lesions are more likely to lead to instability and metastatic tumors and chordomas are more likely to require stabilization than meningiomas and schwann cell tumors [3] 

### 8.2. Options to Stabilize the CVJ

Although anterior approaches are theoretically an option, they are associated with significant morbidity, which is why posterior approaches have been favored in most of the large studies published to date [3,4]. Although there is one approach, the surgical techniques used are heterogenous and range from the implantation of screws, rods and plates to wires, hocks, bands and grafting. Dual approaches consisting of primary posterior stabilization followed by anterior tumor removal have been shown to be feasible in both adult and pediatric patients [132].

### 8.3. Technical Aspects

Tailored preoperative planning is very important, especially for large tumors that spread laterally or in cases in which aggressive bone removal is indicated and should include complementary posterior instrumentation. In cases where stabilization is required, it can be performed in one operation or as a two-step approach (which may be more likely as many tumors are approached anteriorly for resection). Delayed instrumentation is an option for patients requiring radiation therapy [133].

Fiberoptic intubation should be used to avoid the manipulation of the CVJ. The length of the instrumentation depends on the tumor extension and the decompression (laminectomy, facetectomy, etc.) of the spinal cord and should extend two segments beneath the level of the tumor. 

While aids such as intraoperative imaging and neuronavigation play an important role during resection, especially for endoscopic approaches, their role in the stabilization of this area is limited and the literature is scarce [134]. Similar to their use in resection, printed 3D models of bony structures could be used in selected cases [107,135]. Intraoperative monitoring (IOM) is routinely used during resection as well as stabilization, especially in cases in which both operations are performed in one session [107].

## 9. Limitations and Future Research

Although this review provides an overview of the CVJ anatomy and biomechanical implications and summarizes the recent literature regarding the tumor-associated CVJ instability, some limitations have to be mentioned. The major limitation, intrinsic to the study design, is the fact that the nonsystematic nature of this review is prone to selection bias, which is related to the lack of rigorous eligibility criteria. Moreover, a narrative review is not able to grant a qualitative evaluation of the included studies, and therefore it is not able to provide proof of the evidence level of the recommendations reported. However, we followed the SANRA recommendation to improve the methodology integrity. Despite some general indications for the selection of the optimal surgical approach, according to the anatomical location of the lesion and recommendations for surgical fixation, prospective and/or multicentric studies are missing in this field. Future high-quality studies are needed to identify new evidence on this topic, especially to better find instability criteria for those patients requiring a surgical fixation after oncological resection in the CVJ area.

## 10. Conclusions

Tumors of the CVJ still remain a surgical challenge because of the rarity and sparse clinical series on postoperative instability after CVJ tumor resection. Anterior—both open and endoscopic—as well as posterior and posterolateral approaches are available in the surgical armamentarium to deal with CVJ lesions. The choice of surgical approach with regard to the CVJ is largely dependent on the tumor location and degree of invasion of the surrounding anatomical structures. Careful preoperative planning, such as using dynamic XR, CT or MR scans, could support decision making in terms of undertaking anterior or posterior surgical approaches when removing CVJ tumors. According to the current clinical and biomechanical evidence, a disruption of the C1 anterior arch and transverse ligament—especially when undertaking anterior approaches—and the entity of resection or tumor invasions of occipital condyles are factors that are related to postoperative instability and require a CVJ fixation. Moreover, a critical analysis of the need for fixation and of the materials remain essential as most of the CVJ tumors may undergo postoperative adjuvant photon therapy or high-particle such as proton or carbon ion radiation therapy. Future studies, even multicentric and collaborative, are needed to further elucidate the need for fixation in the perioperative period.

## Figures and Tables

**Figure 1 diagnostics-13-01502-f001:**
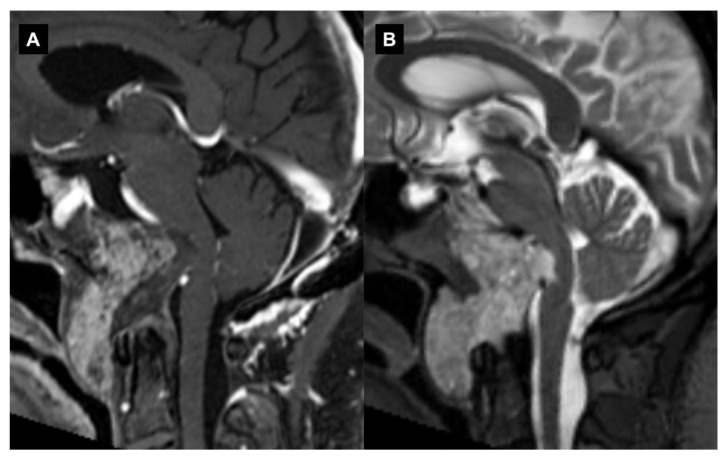
Craniovertebral junction chordoma. Sagittal (**A**) T1-weighted image after contrast and (**B**) T2-weighted image depicting a large chordoma invading the rhinopharynx and extending into the premedullary cistern.

**Figure 2 diagnostics-13-01502-f002:**
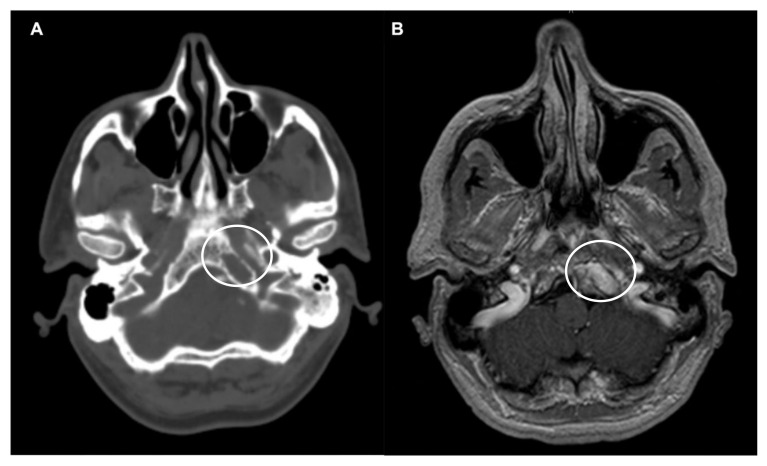
Lower clivus solitary plasmacytoma. (**A**) Axial CT scan showing a lytic expansile lesion at the left lower clivus (circle). (**B**) Axial T1-weighted after contrast injection image showing a heterogenous and hyperintense lesion (circle). A minimally invasive endoscopic endonasal biopsy disclosed the plasmacytoma.

**Figure 3 diagnostics-13-01502-f003:**
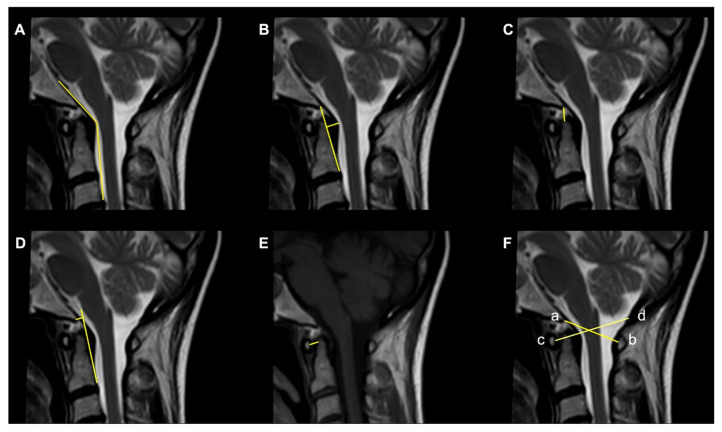
Commonly used radiological parameters to predict CVJ instability. (**A**) Clivoaxial Angle (CXA). (**B**) Grabb–Oakes line. (**C**) Basion–Dens Interval (BDI). (**D**) Basion–Axial Interval (BAI). (**E**) Atlantodental Interval (ADI). (**F**) Powers ratio: ab/cd.

**Figure 4 diagnostics-13-01502-f004:**
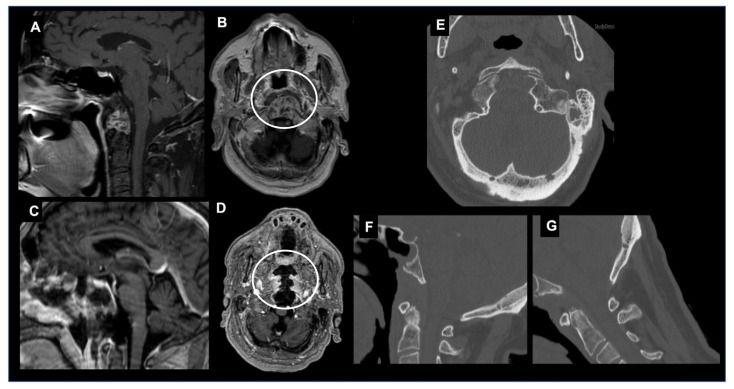
Endoscopic endonasal approach (EEA) to a CVJ chordoma. Sagittal (**A**) and axial (**B**) T1-weighted images after contrast injection showing a craniovertebral junction chordoma invading the C1 anterior arch, transverse ligament and tip of the odontoid. The patient underwent a gross total removal through an EEA. Postoperative sagittal (**C**) and axial (**D**) T1-weighted images after contrast injection confirmed the entity of resection and the integrity of C1-C2 joint. (**E**) Axial CT scan showing the occipital condyle integrity >90%. Dynamic cervical spine CT scans in maximal extension (**F**) and flexion (**G**) showing no abnormal movements and excluding any postoperative CVJ instability.

**Figure 5 diagnostics-13-01502-f005:**
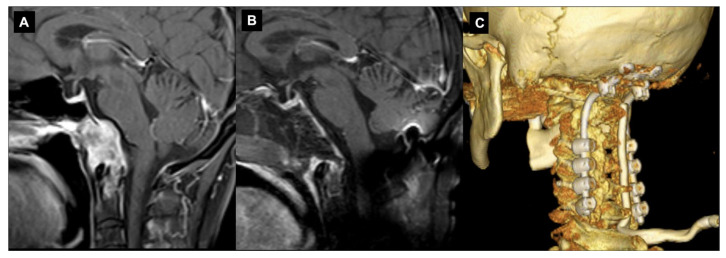
Endoscopic endonasal approach to CVJ chordoma and occipitocervical fixation. Sagittal preoperative (**A**) and postoperative (**B**) T1-weighted MR images after contrast injection showing the chordoma infiltration of C0-C1-C2 complex joint and a gross total resection. In the same surgical setting, an occipitocervical fixation was performed. A 3D reconstruction of the postoperative CT (**C**).

**Figure 6 diagnostics-13-01502-f006:**
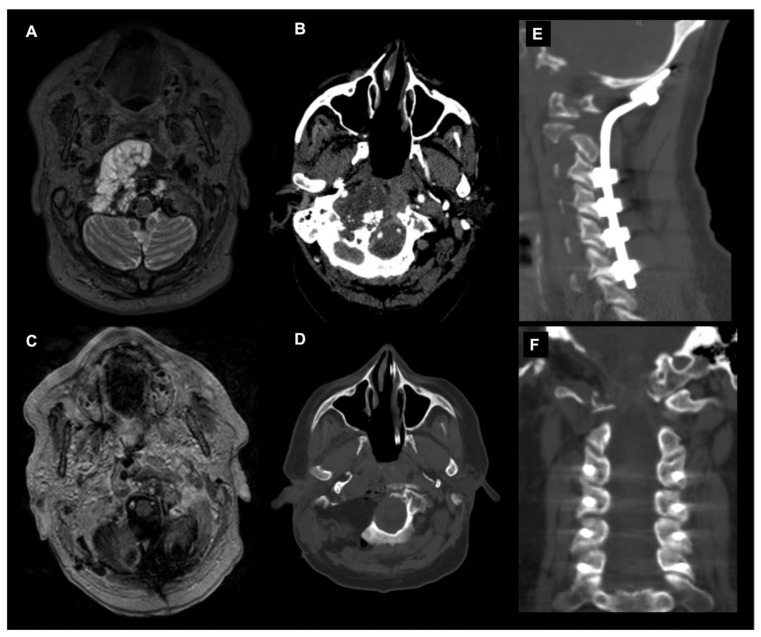
Combined endoscopic endonasal approach and far-lateral transcondylar and petro-occipital trans-sigmoid approach to recurrent CVJ chordoma and subsequent occipitocervical fixation. Axial T2-weighted MR image (**A**) and angio CT scan (**B**) showing a recurrent predominantly right craniovertebral junction chordoma. The chordoma infiltrates the rhinopharynx, C0-C1-C2 joint complex and the entire right occipital condyle. A combined endoscopic endonasal approach associated with far-lateral transcondylar and petro-occipital trans-sigmoid approach has been performed. (**C**) Axial postoperative T1-weighted after contrast injection image and (**D**) noncontrast CT scan disclosed a gross total resection with the destruction of the right clival–atlo–axial joint. An occipital-cervical fixation was therefore postoperatively planned and performed (**E**,**F**).

**Table 1 diagnostics-13-01502-t001:** Advantages and disadvantages for each surgical approach.

Surgical Approach	Advantages	Disadvantages
Transoral	Provide direct anterior access to the CJV from the lower portion of the clivus to C3 Furnish a safe trajectory for extradural midline lesions, avoiding traction and/or manipulation of critical anatomical structures (e.g., cranial nerves, vertebral arteries, brainstem) Permit an excellent decompression of the ventral medulla and upper cervical spinal cord, especially in irreducible ventral pathology	High risk of morbidity including swallowing and respiratory complication, CSF leakage and meningitis in case of intradural pathology Invasive and destructive approach for the surrounding structures (e.g., soft palate and oropharyngeal mucosa or bony structures in case a wider exposure is needed); this instance can be reduced with the endoscopic transoral approach.
Endoscopic Endonasal	Provide a direct anterior access to the CJV from the clivus to the odontoid process Furnish a safe trajectory for extradural midline lesions, avoiding traction and/or manipulation of critical anatomical structures (e.g., cranial nerves, vertebral arteries, brainstem) Minimally invasive approach that reduces the mortality and morbidity related to the standard transoral approach	Less exposure in the sagittal plane, especially below the axis, compared to the transoral approach Risk of CSF leakage and infection although less frequent than with the transoral approach
Posterior	Provide a safer surgical corridor for intradural tumors compared to anterior approaches in terms of CSF leakage and infections Extreme versatile approach to treat several types of dorsal lesions (e.g., meningiomas, schwannomas, intramedullary tumors) Capability to perform posterior fixation procedure within the same surgical time	Risk of neurovascular injury during dissection procedures Risk of postoperative cervical pain Limited access for the resection of lesions extending into intradural and extradural compartments
Posterolateral	Provide a safer surgical corridor for intradural tumors compared to anterior approaches in terms of CSF leakage and infections Extreme versatile approach to treat several types of dorsal lesions (e.g., meningiomas, schwannomas, intramedullary tumors) Capability to perform posterior fixation procedure within the same surgical time Allows the resection of lesion extending into intradural and extradural compartments	Higher risk of neurovascular injury during surgical exposure compared to posterior approach More challenging compared to the posterior approach, requires adequate surgical expertise

## Data Availability

No new data were created or analyzed in this study. Data sharing is not applicable to this article.

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
