# Peer review of "Craniovertebral Junction Instability after Oncological Resection: A Narrative Review"

_diagnostics, 2023, doi:10.3390/diagnostics13081502_

Round 1

Reviewer 1 Report

I had the pleasure to read the manuscript "Craniovertebral junction instability after oncological resection: a narrative review" submitted by Ottenhausen et al.

In my opinion it can be considered for publication after major revision.

Find below my recommendations:

1. There are a lot of typos and the English language does flow smoothly. I.E. row 38-39: "allowing allowed the development of specific human eye-hand coordination movements" does not make sense.

In addition, row 166 "(ROM - Range Of Movement)" it is instead Range of Motion.

2. The anatomy part is too long and needs to be shorten.

3. I would add a table with a list of the surgical approaches including pros and cons of each of them.

Reviewer 2 Report

Reviewer Comments

Thank you very much for the opportunity to review the manuscript submission entitled: Craniovertebral junction instability after oncological resection: a narrative review. The aim of the present review is to summarize the anatomy, biomechanics, and pathology of the craniovertebral junction along with the description of available surgical approaches and consideration of joint instability after craniovertebral tumor resection. The review is interesting, and it has a relevant rationale, however, some limitations and constructive comments are pointed out below:

Specific comments

Abstract:

·      Include MeSH terms as keywords.

·      What are the main take-home messages from the review?

·      what are the implications for future research, Mention it in the abstract.

Main text :

·      Describe the rationale for the review in the context of what is already known.

·      Specify the key question identified for the review topic.

·      Specify the process for identifying the literature search (eg, years considered, language, publication status, study design, and databases of coverage).

·      Mention the limitations and quality of the research reviewed and the need for future research.

·      Provide an overall interpretation of the narrative review in the context of clinical practice.

Round 2

Reviewer 1 Report

Accepted in the revised form.

Reviewer 2 Report

The authors have addressed all the comments. The manuscript can be accepted in its current form.